# How global DNA unwinding causes non-uniform stress distribution and melting of DNA

**Korbinian Liebl**, **Martin Zacharias** *

Physics Department T38, Technical University of Munich, Garching, Germany

* zacharias@tum.de

**Data Availability Statement:** All relevant data are within the paper and its Supporting Information files.

## Abstract

DNA unwinding is an important process that controls binding of proteins, gene expression and melting of double-stranded DNA. In a series of all-atom MD simulations on two DNA molecules containing a transcription start TATA-box sequence we demonstrate that application of a global restraint on the DNA twisting dramatically changes the coupling between helical parameters and the distribution of deformation energy along the sequence. Whereas only short range nearest-neighbor coupling is observed in the relaxed case, long-range coupling is induced in the globally restrained case. With increased overall unwinding the elastic deformation energy is strongly non-uniformly distributed resulting ultimately in a local melting transition of only the TATA box segment during the simulations. The deformation energy tends to be stored more in cytidine/guanine rich regions associated with a change in conformational substate distribution. Upon TATA box melting the deformation energy is largely absorbed by the melting bubble with the rest of the sequences relaxing back to near B-form. The simulations allow us to characterize the structural changes and the propagation of the elastic energy but also to calculate the associated free energy change upon DNA unwinding up to DNA melting. Finally, we design an Ising model for predicting the local melting transition based on empirical parameters. The direct comparison with the atomistic MD simulations indicates a remarkably good agreement for the predicted necessary torsional stress to induce a melting transition, for the position and length of the melted region and for the calculated associated free energy change between both approaches.

## Introduction

In living cells DNA is under permanent torsional stress due to supercoiling and packing by proteins. The response to torsional stress is sequence-dependent and of high relevance for many biological processes. It can facilitate protein binding but can also alter DNA's global topology and influence properties at distant sites to induce protein association or damage recognition [1, 2, 3, 4, 5]. It is well established that the stability and flexibility of linear unrestrained duplex DNA is largely determined by local nearest neighbor effects. The stability or thermal melting of a double-stranded DNA oligonucleotide of any sequence can be estimated

**Funding:** MZ received financial support by the DFG
(grant SFB749/A5; www.dfg.de). The funder played
no role in the study design, data collection and
analysis, decision to publish or preparation of the
manuscript.

**Competing interests:** The authors have declared
that no competing interests exist.

quite accurately from an empirical nearest-neighbor model that just includes parameters for
base pairing as well as nearest-neighbor stacking parameters [6, 7]. The conformational flexi-
bility of a DNA duplex near equilibrium can be described well by a (sequence-dependent) stiff-
ness matrix describing the equilibrium fluctuations of the helical parameters (e.g. twist, rise,
roll etc.) and how they are coupled to nearest neighbors [8, 9, 10, 11, 12]. Hence, a twist fluctu-
ation at a given base pair step in a linear DNA duplex has a significant influence on the helical
parameters of the next base pair step but little (or no) influence on steps at farer distance. The
nearest-neighbor stiffness of DNA can be extracted from Molecular Dynamics (MD) simula-
tions and allows one to calculate global flexibility parameters such as bending, twisting,
stretching as well as twist-stretch coupling in good agreement with experiment [8, 9, 10, 11,
13].

However, as pointed out already by Benham [1, 14, 15] the situation changes completely if a
global constraint is included such as circularization of the duplex or binding of linear DNA to
fixed anchors such as DNA packing complexes. In such situations a long range coupling due
to global competition among local conformational states influences the flexibility and stability
of the duplex. Hence, now a local change in a helical variable such as twist (e.g. due to melting)
can couple with the relaxation of an over- or under-twisted segment at a large distance. The
long-range coupling has important consequences for the dynamics of supercoiled DNA and in
particular for the local denaturation that can be induced by not only forming a circular DNA
but by applying in addition a global unwinding stress [1, 14, 16, 17]. Unwinding in supercoiled
DNA can result in local strand dissociation of the DNA, which is a central event for DNA rep-
lication and transcription. Ising-type models have been introduced as a theoretical concept to
calculate and predict the effect of torsional stress on the destabilization and denaturation of
supercoiled DNA [14, 15, 16, 17]. These models are based on empirical parameters for the
local stiffness of dsDNA and melted ssDNA as well as the sequence-dependent local dsDNA
stability. It allows one to rapidly calculate duplex destabilization regions in long DNA
sequences [16, 17] and these regions have been shown to often overlap with promoter regions
and transcription start sites [5, 15, 18, 19]. Several subsequent studies also specifically empha-
sized a critical influence of unwinding in supercoiled DNA in particular on transcription initi-
ation [15, 20, 21, 22]

The effect of torsional stress on globally constrained DNA can also be investigated using
single molecule torque measurements [4, 23, 24, 25, 26, 27, 28]. It has been used successfully to
study the twist flexibility and twist-stretch coupling of DNA and RNA and to characterize also
the torsion-induced melting/denaturation of DNA [27, 29]. Interestingly, strong unwinding of
DNA can result in left-handed twisting (L-DNA) of melted regions and even formation of
DNA structures characteristic for left-handed conformations such a Z-DNA [23, 24, 30].

Nevertheless, even single-molecule experiments provide only the measurement of effects
on the whole molecule with limited insight into the response of individual base-pairs or seg-
ments of the DNA. In addition to different experimental approaches, DNA unwinding has
also been studied employing Molecular Dynamics (MD) simulations [22, 31, 32]. Most of
these studies are, however, limited to an untwisting regime covering the elastic response (qua-
dratic relation between deformation and energy) and its sequence dependence. Only recently
also untwisting beyond the elastic response towards local melting has been systematically stud-
ied on small DNA fragments [18, 33] or circular DNA [34, 35]. However, the molecular mech-
anism how torsional stress propagates and is distributed through the DNA sequence is still not
well understood. For example, in relaxed DNA the mean elastic deformation energy is evenly
distributed in each helical parameter that describes the DNA conformation. The application of
a global torsional stress can change this scenario such that certain degrees of freedom deform
strongly and absorb more elastic energy than others. The redistribution of elastic energy

eventually promotes local DNA melting. The molecular mechanism of such mechanically induced phase transitions is also not fully understood.

In the present study, we employ explicit-solvent all-atom MD-simulations to unwind two 50-bp long double-stranded (ds)DNA molecules using an added torque restraining potential. The setup is similar to experimental single-molecule torque experiments. Both DNAs contain a transcription start like sequence (TATA box, rich of adenine and thymine nucleo-bases) embedded in different sequence contexts. For both cases our step-wise unwinding results eventually in local melting of just the TATA box (no melting of other regions). Furthermore, the transition is observed at the expected supercoiling density for which local melting is also observed in experiments. In addition to directly follow the structural details along the whole DNA sequence, the simulations allow us also to calculate the free energy change associated with unwinding and to ultimately induce local melting of the DNA.

Furthermore, from the simulations we have calculated the absorbance of elastic energy for each base pair step based on an elastic model for DNA deformation in helical variables and obtain a discontinuous pattern. Upon torsional stress the deformation energy tends to be stored more in cytidine (C)/ guanine (G) rich segments (associated with enhanced transitions from the BII to the BI substates), while the TATA-boxes remain rather relaxed. This character changes abruptly upon phase transition, where we observe local melting of (only) the TATA-Box for both sequences and the deformation energy absorbed by the denaturation bubble in the TATA-Box, whereas other sequences relax again back to near B-form.

To relate the simulation results to experimental parameters we designed an Ising model similar to previous approaches [1, 6, 14, 15, 17] for describing both the average free energy of the constrained system as well as the melting probability under torsional stress. The model is entirely based on empirical nearest neighbor base pair/stacking parameters as well as DNA twist elasticities. The direct comparison with the MD simulation results indicates a remarkably good agreement for the necessary torsional stress to induce a melting transition and the position of the melted segment. In addition, the shape of the free energy vs. supercoiling density agree excellently in the elastic regime and differ only slightly in the phase stability that likely indicates a small over-stabilization of the dsDNA by the MD force field description. Besides of new insights into the molecular mechanism of torsion-stress induced DNA melting our study indicates that MD simulations are not only useful to study equilibrium properties of DNA but also to investigate large-scale transitions in DNA induced by global constraints or external forces.

## Materials and methods

### Simulation setup

MD simulations were performed on two 50 bp-long dsDNA sequences, one (AT) with approximately 50% randomly distributed A/T content and a near central TATA box sequence starting at base pair 17 (Table 1). The second sequence contains the TATA box at the same position but otherwise consists of only C/G nucleotides in a random sequence. The sequences were processed with the xleap-tool of the Amber16 package [36]. Each system was neutralized by addition of sodium ions and solvated with the TIP3P water-model [37], whereby we used a

**Table 1. Simulated sequences.** TATA-Box location is highlighted in bold.

| Sequence | Label |
|---|---|
| 5′ − cgcgcatgaactgcag**ttatat**ggacctcgatgcggcgtacagtacgcgc−3′ | AT |
| 5′ − cgcgcgcgcgcggccg**ttatat**gggccgcgcggcggcgcgccgcgcgcgc−3′ | GC |

minimum distance of 12 Å between DNA and a rectangular box boundary. All bonds involving hydrogens were constrained to the optimal length using Shake [36]. The DNA structures are initially aligned along the z-axis of the box. We used the most advanced bsc1 force-field for the DNA-parameters [10]. The solvated systems were energy-minimized in 5000 steps using the steepest-descent method. Subsequently, we equilibrated the systems to 300 K in three steps while including positional restraint on DNA's non-hydrogen atoms. The restraints were reduced from initially 25 kcal/(molÅ2) to 0.5 kcal/(molÅ2) in five simulations at 300 K and a pressure of 1 bar. The resulting structures served as the starting structures for production runs of at least 500 ns, where we applied cylindrical restraints with a small force constant of 0.1 kcal/(molÅ2) on the two terminal base pairs of both ends with respect to the z-axis. In case of cylindrical restraints only distances from the z-axis are restrained allowing full rotational freedom (twisting) and full freedom in azimuthal (z) direction. Cylindrical restraints prevent significant overall rotation of the DNA helix axis and limit the overall bending. From an experimental single molecule view this represents the case where supercoiling (strong bending) of the helix is reduced (typically by applying a small stretching force to keep the DNA aligned in a single molecule experiments). From these simulations we inferred the stiffness-matrices for the two DNA sequences (see below). In addition, the twist persistence length of the dsDNAs was calculated using $P = \frac{Contour \cdot k_{tw}}{k_B T}$. The mean contour-length of a DNA segment was obtained from the sum of the base pair step distances and the stiffness $k_{tw}$ along the contour as the inverse of the observed twist variance of the final base pair of the segment with respect to the starting base pair.

The final structures from the above production simulations served also as input for Umbrella Sampling simulations. In the Umbrella Sampling (US) simulations we used a dihedral angle as reaction coordinate. The dihedral angle is defined as $\xi = \angle$ (:4@C1': 97@C1': 53@C1': 47@C1'), representing a rotation of the 47th base pair with respect to the 4th base pair. During the simulations, the reaction coordinate was biased by a harmonic potential: $V = k \cdot (\xi - \xi_0)^2$, with k = 0.0122kcal/(mol · deg$^2$) (= 40kcal/(mol · radians$^2$)). The reference-value $\xi_0$ was changed in steps of 10$^o$ per US interval (window). Simulations of every window were performed for at least 100 ns and the first 50 ns have been skipped in the analysis. As shown in S1, S2 Figs (in S1 File), this was sufficient to achieve converged free energy profiles. All simulations were performed with the pmemd.cuda module of Amber16. Throughout the manuscript we indicate the degree of unwinding by just the magnitude of supercoiling density $\sigma$ (positive number) and do not use a negative $\sigma$ as frequently used to indicate unwinding in circularly closed DNA. It indicates the number of unwinding turns relative to the equilibrium helical turns in the DNA.

## Calculation of stiffness and covariance matrices

Stiffness matrices for both sequences were obtained by inversion of the covariance matrix: $K = k_B T \cdot C^{-1}$, whereby we used the twist, roll and tilt angles of all 43 central base-pair steps. The helical parameters were calculated from the unrestrained MD-trajectories using the program Curves+ [38]. We obtained a twist persistence length of 110.8 nm and 119.9 nm from the stiffness matrix for the AT- and GC-sequences, respectively. The computed stiffness matrices are depicted in S3, S4 Figs (in S1 File). The corresponding covariance matrices are illustrated in S5-S8 Figs in S1 File. Little covariation between helical parameters beyond nearest neighbors can be observed for the simulations of the unrestrained systems (S5, S7 Figs). This changes dramatically upon restraining the global twist of the systems and reflects one of our considerations in the following paragraphs: Global restraints on the total twist evoke significant correlations between distant sequences of the DNA molecules.

## Results and discussion

### DNA as a set of harmonic oscillators

Close to the equilibrium structure the energy landscape of dsDNA can be well described as a harmonic system in terms of intra- and inter base pair conformational parameters [8–10]. Hence, in terms of the helical parameters the system is described by a quadratic deformation energy surface. For an unrestrained system one expects at equilibrium on average an equipartitioning of deformation energy E. On a simple mechanical system we first demonstrate how the equipartition of deformation energy changes upon introducing a global restraint (such as an external restraint on the twist in case of DNA). We consider two harmonic oscillators, for simplicity we neglect coupling terms as it does not effect the general mechanism outlined here:

$$E = k_1 \cdot (q_1 - q_1^0)^2 + k_2 \cdot (q_2 - q_2^0)^2, \tag{1}$$

whereby the individual deformation energies are given by $E_1 = k_1 \cdot (q_1 - q_1^0)^2$ and $E_2 = k_2 \cdot (q_2 - q_2^0)^2$. The expected energy is determined by the virial theorem:

$$\langle q_i \frac{\partial E}{\partial q_j} \rangle = \delta_{ij} k_B T \tag{2}$$

When freely exposed to a thermal heat bath, both oscillators absorb on average the same amount of deformation energy,
$\langle q_1 \frac{\partial E}{\partial q_1} \rangle = \langle 2k_1 q_1 \cdot (q_1 - q_1^0) \rangle = \langle 2k_1 \cdot (q_1 - q_1^0)^2 \rangle = \langle 2 \cdot E_1 \rangle = \langle 2 \cdot E_2 \rangle = k_B T.$

However, if we impose a global restraint on the system (e.g. by stretching or winding of DNA), $E^{res} = k_R \cdot (L - L_0)^2$, all parts of the system are coupled through the geometric condition $q_1 + q_2 = L$. As a consequence of the virial theorem, thermal energy is then no longer equipartitioned between the individual oscillators but between coupled modes:

$$k_B T = \langle 2 \cdot k_i q_i \cdot (q_i - q_i^0) + 2 \cdot k_R q_i \cdot (q_i + q_j - L_0) \rangle \tag{3}$$

Thus, equipartitioning of deformation energy, $\langle E_1 \rangle = \langle E_2 \rangle$, is no longer obligatory. Hence, global restraints can induce correlations also between distant sites. Intriguingly, this phenomenon is also revealed by our MD simulations on DNA upon inclusion of a global restraint on the twist. In the case of unrestrained DNA the twist variation at a given base pair step shows only significant covariation with nearest neighbors (non-diagonal elements in the covariance matrix away from the diagonal are close to zero) (Fig 1). However, significant non-nearest neighbor correlations arise upon addition of a global unwinding restraint during the simulations (will be further discussed in the paragraph on Molecular Dynamics simulations of DNA unwinding, see Fig 1). This can be understood qualitatively since a global coupling allows, for example, to relax a local deformation (in twist) by an appropriate opposite twist change at a distance to fulfill the global restraint. Indeed, such long-range correlations in DNA mini-circle simulations have been reported by Sutthibutpong and coworkers [35]. In our simulations we continuously increase the torsional stress on linear fragments which allows us to study the changes of twist and the distribution of conformational substates along the DNA up to the melting regime and the associated re-distribution of elastic energy (see next paragraphs).

### Local bimodality in the twist distribution of dsDNA

A prominent conformational polymorphism in DNA is due to two different combinations of the $\epsilon$ and $\zeta$ dihedral angles in nucleotides adopting either the canonical BI ($\epsilon/\zeta$ in the trans/gauche regime) or BII configuration ($\epsilon/\zeta$ in the gauche-/trans regime) [8, 12, 32, 39, 40]. These

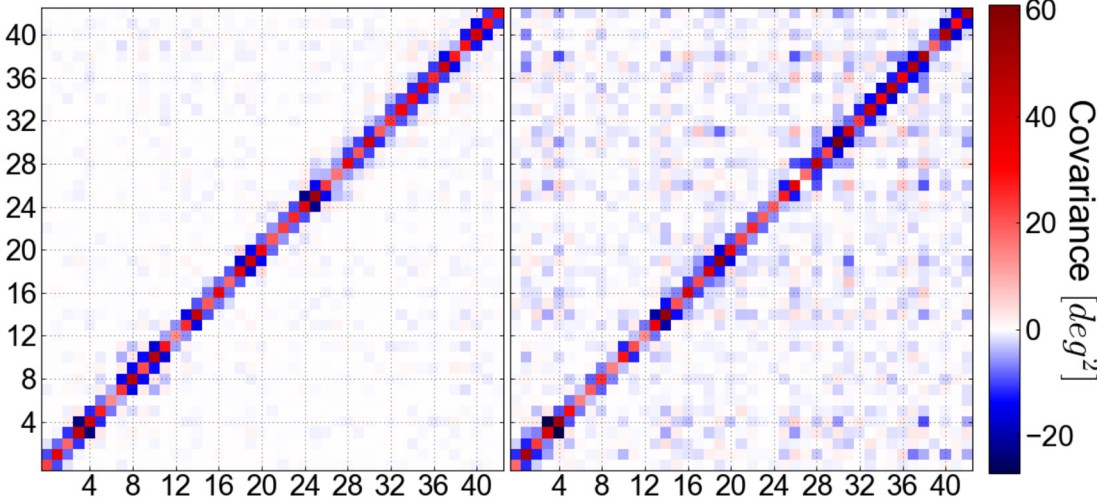

**Fig 1. Covariance matrix of the twist fluctuations obtained for the central 43 base-pair steps during MD-simulations (at-sequence).** In the unrestrained case (left panel), coupling exists only between nearest neighbor sites. Upon global unwinding (right panel, $\sigma = 0.067$), also distant sites become coupled to a significant extent.

local states can also change the equilibrium twist and twist flexibility and can trigger small deviations from the previously discussed harmonic behavior for several modes of the dsDNA (Fig 2A). Typically, a local BII state is associated with a slightly larger equilibrium twist and a more narrow distribution compared to a BI state [12, 40]. Hence, upon global unwinding the torsional stress may relax not only by small changes of the local twist from the equilibrium but also by switching from BII to BI states upon unwinding. It has also been pointed out, that the population of the backbone states is sequence dependent. This point has recently been taken up by Reymer et al. (2017) who have proposed that specific sequences act as 'twist capacitors' as a consequence of the backbone bimodality. In their study, the authors demonstrate a sequence dependence in relative changes in twist, but did not relate it to local stiffnesses [32].

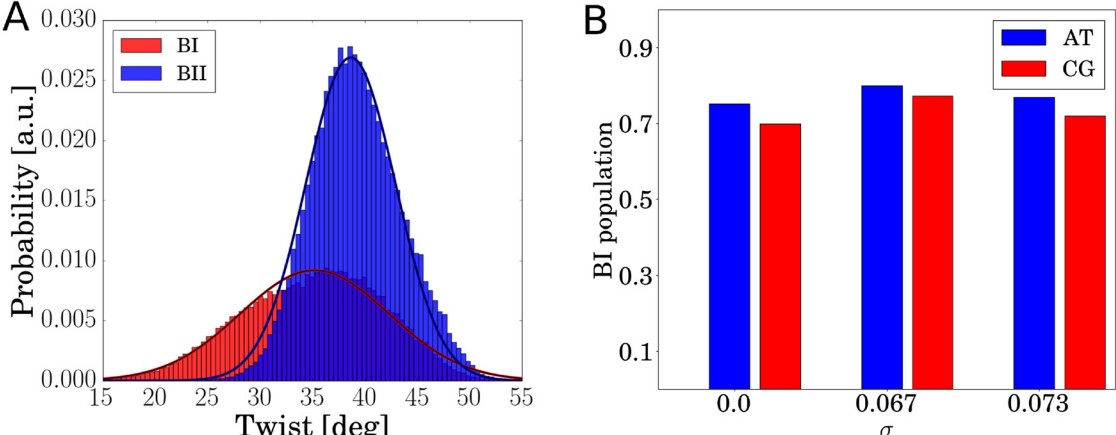

**Fig 2.** (A) Twist-distribution sampled for a CpG nucleotide step during unrestrained MD simulation. The probability distribution is bimodal due to the sampling of the two possible backbone substates termed BI and BII. BI states are associated with an on average lower twist and have higher variance than BII states for the associated CG base pair step. (B) Changes in the overall BI population at different levels of global unwinding (for AT and CG steps). At a $\sigma = 0.067$ an increased BI level is observed that relaxes for higher $\sigma$ due to local DNA melting and relaxation of the rest of the dsDNA.

It has therefore remained unclear how much of the deformation energy is absorbed due to backbone polymorphism.

## Structural changes in dsDNA upon global unwinding

We performed Molecular Dynamics (MD) simulations including a torque on the dsDNA termini to continuously unwind two different, 50 bp-long, dsDNA molecules (Fig 3). Note, that both sequences contain an AT-rich segment ('TATA-box', often found at transcription start sites in DNA), with our second sequence containing otherwise only G-C base pairs (termed CG-sequence) and the first sequence also several A-T base pairs (in total almost 50%, therefore termed AT-sequence). It allows us to elucidate the impact of the G-C content on DNA's twist deformability and absorbance of deformation energy. Using a torsion reaction coordinate, similar to previous studies [31, 33], it is possible to unwind the central 43 bp-steps of the DNA oligomers down to local melting by employing the Umbrella Sampling (US) technique (Fig 3). It is important to note, that our sequences are longer than those from most previous MD studies (usually 10-20 bp). For a given supercoiling density (or fractional unwinding: number of unwinding turns per equilibrium turns in DNA) $\sigma$ this allows the DNA to "store" a sufficient number of unwinding turns that can eventually relax due to melting of a DNA segment of several base pair steps. Note, that we indicate the degree of unwinding by just the magnitude of supercoiling density $\sigma$ (positive number) and do not use a negative $\sigma$ as frequently used to indicate unwinding in circularly closed DNA.

With increasing global unwinding we observe an increase in the coupling of helical variables beyond nearest neighbors (Fig 1 and S3-S6 Figs in S1 File).

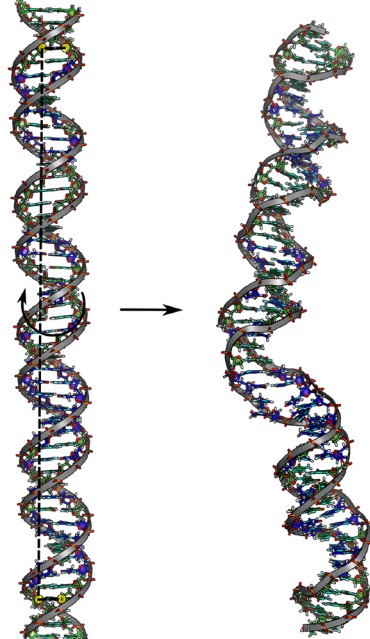

**Fig 3. Setup of the simulation systems (as stick models with the backbone indicated as cartoon).** Yellow circles indicate positions of the C1' atoms (4th and 47th bp), which constitute the torsional reaction coordinate, illustrated as dashed line (unwinding is indicated as curved arrow in the middle of the left panel). Step-wise unwinding eventually leads to melting of the TATA-box segment (located in the lower half of the DNA molecule, right panel). C/G bases are shown in green, A/T bases in blue.

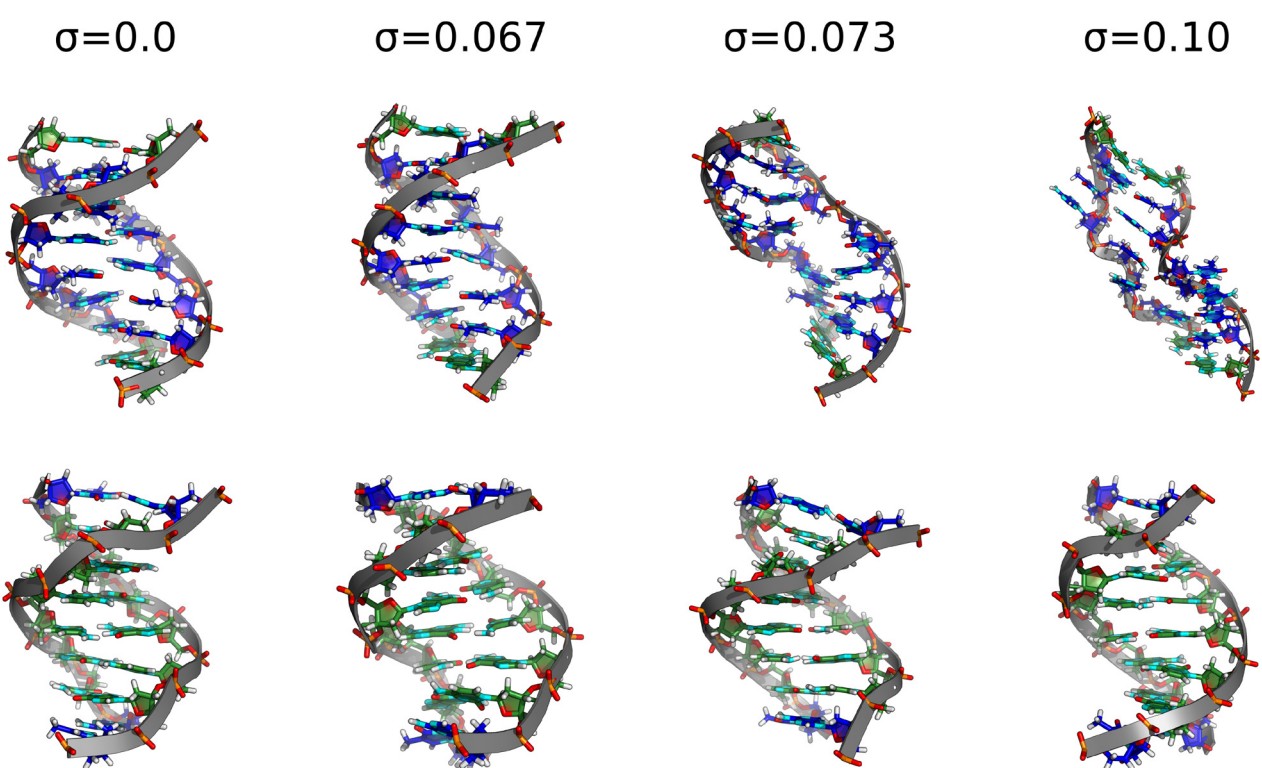

**Fig 4. Snapshots of two selected 8 bp segments of the DNA at different levels of global unwinding (gc-sequence).** Structures of the TATA-box containing segment are shown in the upper series. In the lower panels structures of a distant GC-rich segment are illustrated (same color-coding as in Fig 3).

As discussed already, a global unwinding restraint stabilizes different combinations of twist and other helical variables along the sequence that fulfill the global restraint and thus can enforce increased long-range covariation or co-fluctuations. Since the BI substate is on average associated with a smaller local twist compared to BII increased global unwinding also results in an increase of the BI/BII ratio (Fig 2B). At a global unwinding $\sigma$ of 0.07, base pairs at the TATA box are disrupted and start to melt (Figs 3 and 4). Notably, our simulations show local melting of the TATA-Box (and not other regions) for both sequences. At a global unwinding of $\sigma = 0.067$ most base pair steps along the DNA sequence are significantly undertwisted relative to regular DNA but for $\sigma = 0.073$ (upon melting) the local twist deformation is redistributed such that most of the unwinding is "absorbed" by the TATA box segment and the rest of the DNA relaxes to an average twist closer to the mean twist in relaxed B-DNA (Fig 5, Table 2). Also, the local bending of DNA in the TATA box slightly increases which also relaxes part of the torsional stress (Table 2). Plotting the twist of just the TATA box segment vs. global unwinding reveals three phases: At modest unwinding stress the twist changes gradually (regime I) and upon melting an abrupt large change is observed (regime II, Fig 6, Table 2) followed by a regime III with much steeper twist change compared to regime I. This indicates that in this regime further external unwinding stress is basically entirely "absorbed" by the melted region (with a much smaller twist stiffness than regular dsDNA). It is indeed possible that unwinding of a longer dsDNA (with more "stored" unwinding turns) can result in left-handed DNA structures in the melted region. Several experimental studies indicate that indeed left-handed forms of DNA can occur in strongly supercoiled DNA [23, 24, 30]. Consistent with previous studies on base pair formation and melting [41, 42, 43] we found also an

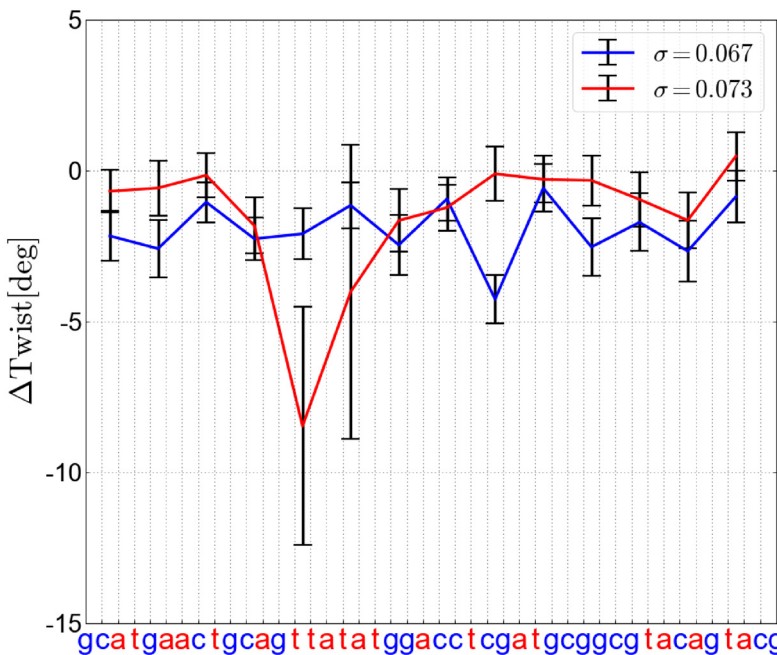

**Fig 5. Average changes in twist at each base pair step (relative to unrestrained DNA, at-sequence) upon global unwinding for a superhelical stress $\sigma$ slightly below melting (blue) and above melting in the TATA box segment (red line).** The average deviations (with respect to fully relaxed unrestrained DNA) and standard deviations were calculated as averages of three adjacent base-pair steps.

asymmetric fraying/breathing tendency of adenine vs. thymine in the melting region. For estimating this tendency the center of the two sugar rings of a base pair was calculated yielding a center point. The distance B of the center point to a reference point in the base (N1 for adenine, N3 for thymine) was then compared to the distance S between the sugar and the center point. If distance S < distance B, the base is counted as frayed, otherwise not. The rationale behind this procedure is that in the frayed state, the base rotates away from the center point of the base pair and hence the sugar is then closer to the center of the base pair. Hereby, we obtain on average increased breathing of thymine bases in the TATA-box upon melting by a factor of 1.2 (at-sequence) and 1.3 (gc-sequence).

It is also interesting to compare the structure of the TATA box segment with other parts of the DNA of similar length that do not melt (Fig 4). Whereas the TATA box segments undergo dramatic structural changes at $\sigma = 0.073$ one observes for other segments mainly a continuous unwinding and coupled shrinking along the helix (due to twist-stretch coupling in DNA that results in shrinking of the DNA upon untwisting [13]) but no structural transitions to a new

**Table 2. Deformation of the TATA-Box under different levels of unwinding, averaged over both sequences.**
Parameters have been obtained by computation of the rigid-body transformation from the C-G base-pair prior to the TATA-Box to the first C-G base-pair after it. Our procedure is explained in chapter 5 in S1 File.

|  | Twist/bp [deg] | Bending/bp [deg] |
|---|---|---|
| $\sigma = 0.0$ | 34.37±1.21 | 2.32±1.24 |
| $\sigma = 0.067$ | 31.88±1.39 | 3.13±1.46 |
| $\sigma = 0.073$ | 21.59±2.79 | 4.45±1.83 |

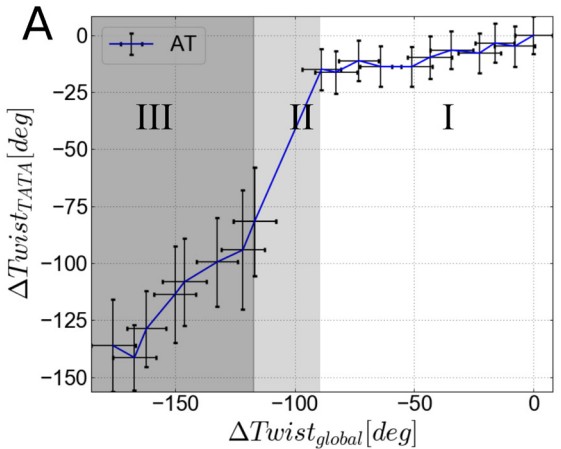
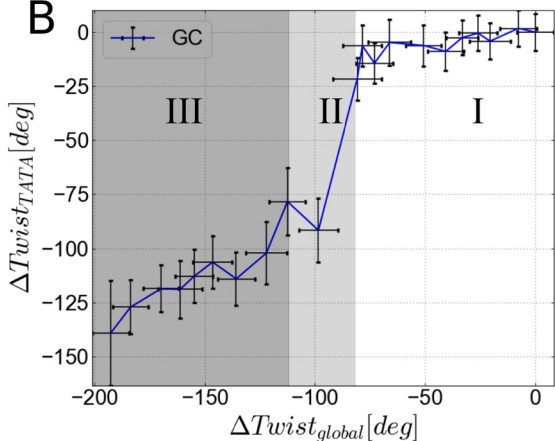

**Fig 6. Change in total twist in the TATA box segment upon global unwinding for the at-sequence (A) and gc-sequence (B).** In the regime I (harmonic regime) a continuous unwinding is observed that changes abruptly during the melting phase (regime II). During this phase-transition global unwinding is largely stored in the TATA-box segment. Further unwinding causes continuous unwinding of the TATA-Box (regime III) with a steeper slope than in regime I (due to the much smaller twist persistence length of melted vs. intact DNA). Global twisting and twisting of the TATA-Box segment have been computed with a protocol described in chapter 5 in S1 File (same as in Table 2).

form of DNA. The melting of the TATA box segment results in a relaxation of other segments (and slight expansion/stretching due to increased local twist, Fig 4).

## Redistribution of DNA elastic energy upon global unwinding

The deformability and conformational flexibility of a dsDNA molecule near equilibrium can be well described by a quadratic model for the deformation energy in terms of the helical DNA conformation parameters (e.g. twist, roll, tilt etc.) [9, 11, 33]. Such a model allows us to calculate the mean deformation energy for each base-pair step based on a harmonic model (details of the model are given in Chapter 2 in S1 File):

$$E(\Delta w) = \frac{1}{2} \Delta w^T \cdot K \cdot \Delta w, \tag{4}$$

where $\Delta w$ denotes deviations of the internal coordinates from the equilibrium values and K the stiffness matrix, obtained as inverse of the covariance matrix of the internal coordinates: $K = k_B T C^{-1}$. Note, that the stiffness matrix and the equilibrium values of the internal coordinates have been determined from extensive unrestrained MD simulations (see Methods and chapter 2,3 in S1 File). Since changes in twist may also couple to bending of the DNA, we selected the twist, roll and tilt variable of each of the 43 central bp-steps as internal coordinates. This form assumes that other helical deformations fully relax according to the corresponding covariation. It is also possible to use a more extended stiffness matrix with more helical parameters but this will not change the redistribution of elastic energy. The selected variables fully determine twisting and local bending of the DNA molecule. Thus, K becomes a 129x129 matrix. The elastic energy for each base pair step included the helical variation and half of each non-diagonal contribution. Not surprisingly, in the absence of external torsional stress we obtain exactly the expected deformation energy-equipartitioning with a mean deformation energy of $\sim \frac{k_B T}{2}$ per degree of freedom (Fig 7, S9, S10 Figs in S1 File). Similarly, we are also able to evaluate the absorption of local deformation energy along the sequence upon application of torsional stress. The amount of global unwinding is quantified by our fractional unwinding coordinate $\sigma$,

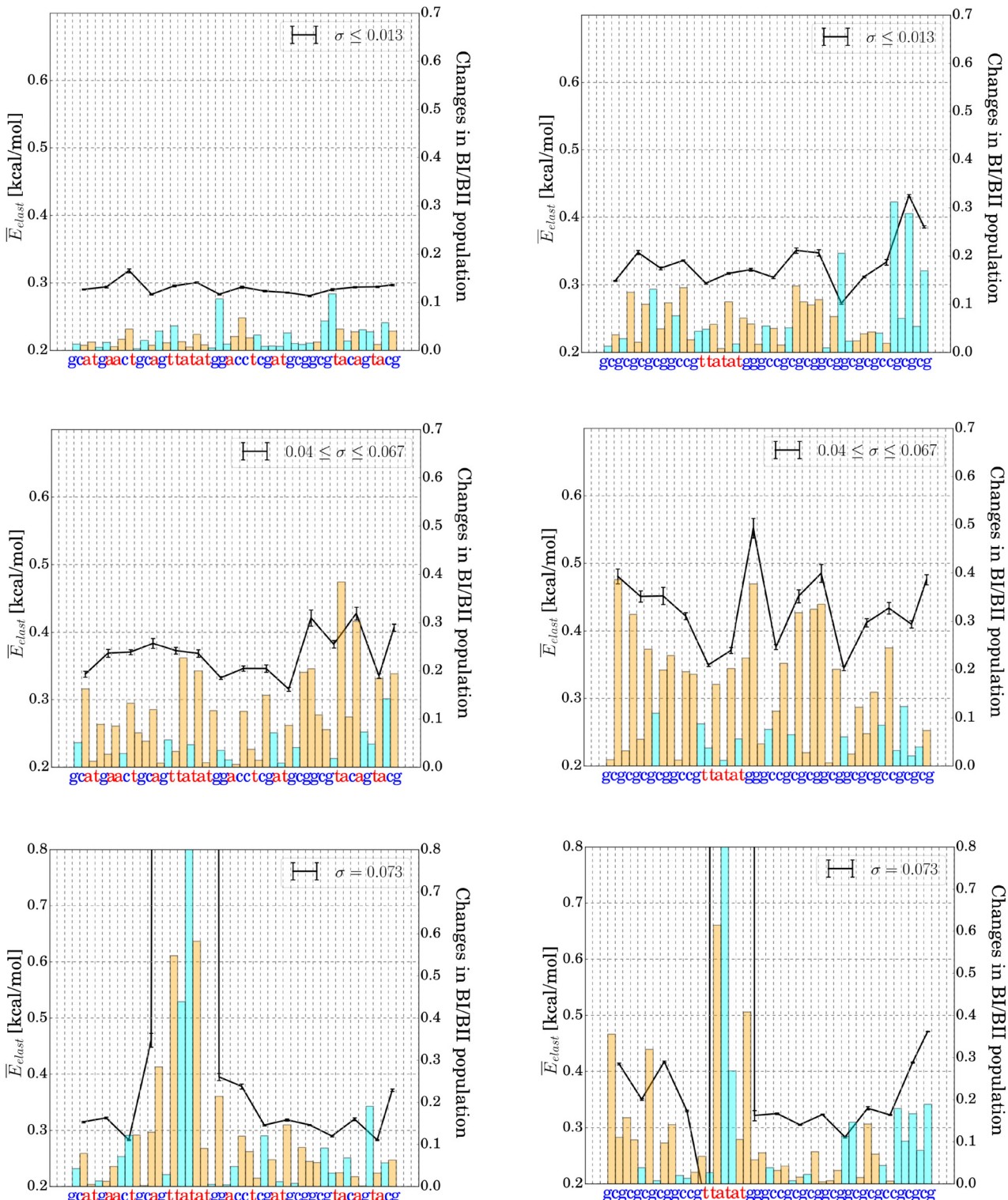

**Fig 7. Average elastic energy per degree of freedom and changes in BI/BII conformational substate population with respect to unrestrained simulations for each of the central 43 bp-steps of the heterogeneous at-sequence (left column) and the gc-sequence (right column).** Each vertical dashed line in the panels corresponds to a base pair (sequence indicated on the x-axis). The black lines indicate the average deformation energy at each base pair step. The cyan bars indicate an increase in BII, orange bars an average increase in BI states relative to the unrestrained simulation. The profiles have been generated for different levels of global DNA unwinding. Upper panels show simulation results of Umbrella windows with low unwinding. Middle panels illustrate moderate unwinding simulations, where the DNA duplex has still remained structurally intact. Bottom panels

reflect simulations at strong unwinding, which results in TATA-Box melting for both sequences. In order to calculate $\bar{E}_{elast}$, we determined deformation energies (Eq 4) for every frame and subsequently calculated averages (over three adjacent base-pair steps). Note that a harmonic energy function is insufficient in the denatured phase, corresponding values therefore only have a qualitative meaning. Error bars have been calculated as standard errors by splitting the simulation into bins of 25 ns.

indicating the reduction of global twist relative regular B-DNA. Already at modest induced unwinding, the profiles along the DNA indicate a non-uniform deformation energy distribution (Fig 7 and S11-S31 Figs in S1 File).

Further increase of global torsional stress results in strong fluctuations of absorbed deformation energy along the sequence. It is possible that on the time scale of our simulations not all relevant states compatible with the global restraint have been sampled. However, on average the stress levels of A/T segments are lower compared to C/G segments and the TATA-boxes are more relaxed compared to the C/G segments that absorb more of the global stress. Segments storing more elastic energy also exhibit significant transitions in the backbone structure (Fig 7 and S11-S31 Figs in S1 File). Since, C/G segments exhibit a higher propensity for adopting a BII state in unrestrained DNA [12, 44, 45, 46] these segments have also a higher capacity towards transitions from BII to BI states (which overall relaxes unwinding stress, see previous paragraphs). Especially, at high supercoiling (before melting) significant increase in the BI propensity is observed along the DNA sequences mostly in the G/C-rich segments (orange bars in Fig 7, see also Fig 8). One can interpret this as a possible protection-mechanism indicating that DNA unwinding may not necessarily lead to high rates of A/T-base-pair breathing/flipping. Eventually, further increasing the torsional stress induces transition to a phase which is characterized by local denaturation (at $\sigma \sim 0.07$). A base pair was counted as denatured upon loss of

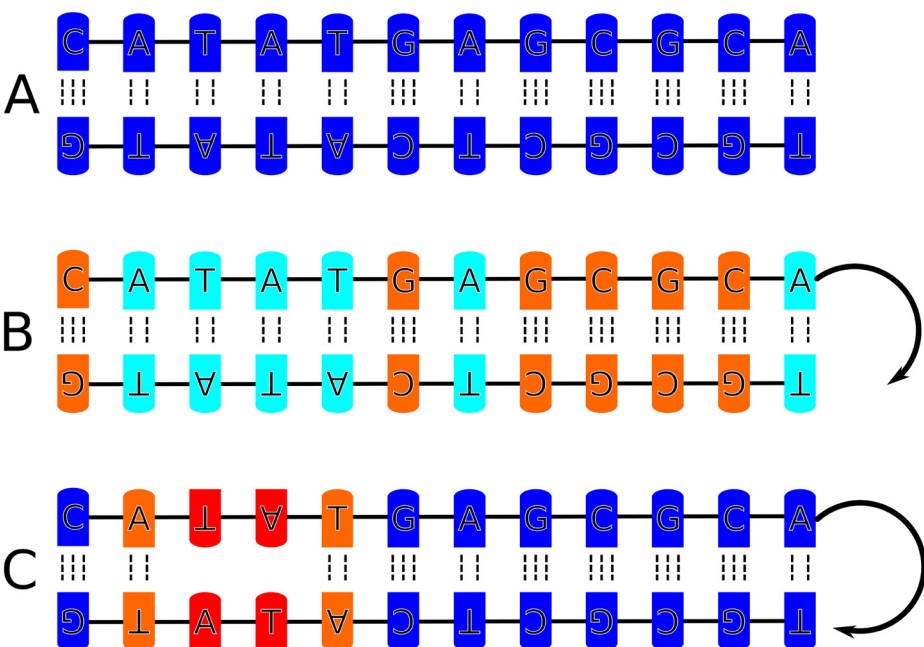

**Fig 8. Display of the sequence-dependent impact of global unwinding.** In the absence of global stress, all sequences undergo equal levels of deformation ($\sim \frac{k_B \cdot T}{2}$, A, highlighted in blue). Global unwinding then causes G-C sequences to absorb most of the stress (red color in B), while promoter like sequences remain rather relaxed (light blue color in B). At strong levels of unwinding, global stress is absorbed through melting of the TATA-Box (red in C), whereby distant sequences are relaxed again (blue in C).

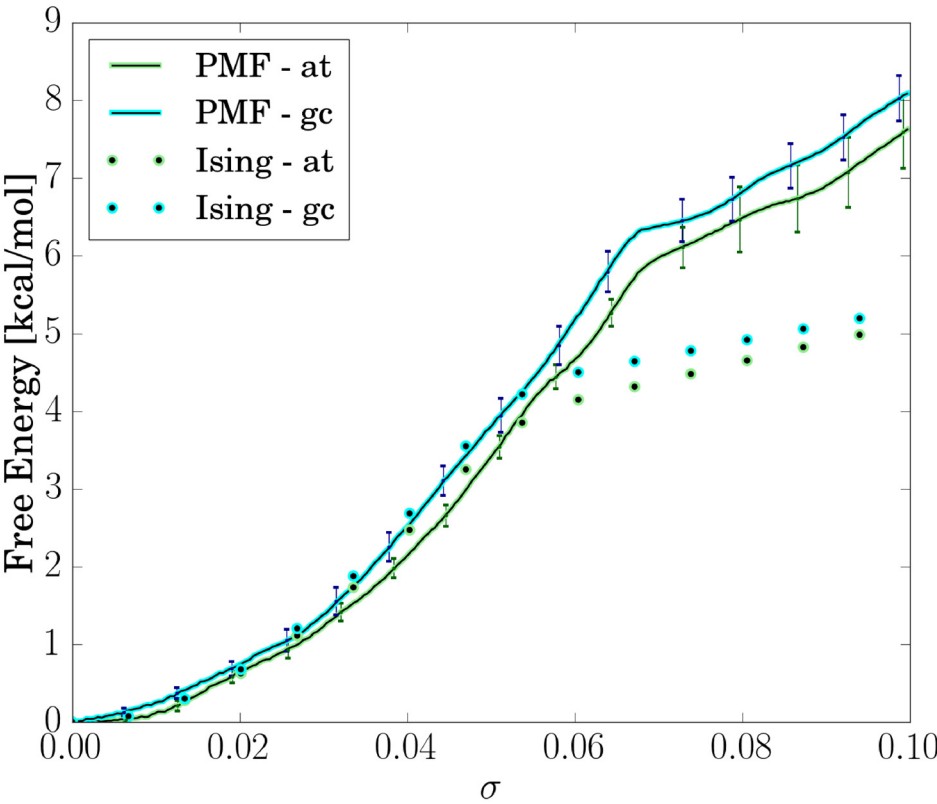

**Fig 9. Free energy profiles for global unwinding of the DNA molecules ($\sigma$) obtained from US simulations (PMF) and calculated by Ising models.** Error bars of the PMFs have been calculated by thermodynamic integration and block-averaging [48].

all Watson-Crick base pair contacts (H-bond donor-acceptor distance > 3 Å) In this phase, most of the deformation energy is absorbed in the denaturation bubble, while all other segments of the DNA are becoming more relaxed again (bottom panel of Figs 7, 5, 6(A) and 6(B)). This is in line with the observed relaxation of twist in the DNA except for the melted region discussed in the previous paragraph. Of course in the melted regime the harmonic elastic model is not anymore valid. Hence, the elastic energies reported in Fig 7 for the melted TATA-region can only be considered as qualitative indicator for absorbing most of the elastic energy of the system. The impact of global unwinding on the deformation energy distribution is presented schematically in Fig 8.

## Calculation of the free energy change associated with global DNA unwinding

The application of the Umbrella Sampling (US) technique to globally unwind the DNA molecules allows us also to extract the associated free energy change by employing the Weighted Histogram Analysis Method (WHAM) [47] (Fig 9). Similar to a previous study on much smaller DNA molecules we find for both sequences an almost quadratic relation between free energy and a relatively wide range of global unwinding $\sigma$ [33]. In line with the results on analysis of the twist in the TATA box segment (see previous paragraphs and Fig 6), we find that at $\sigma$ $\sim 0.07$ the character of the free energy curve changes drastically to a more flat shape with a small slope with respect to further increasing $\sigma$. Interestingly, the transition occurs in both

sequences at similar values of $\sigma$. This regime corresponds to a free energy level of $\sim 6 kcal/mol$ that is required to initiate the TATA-Box melting. In the melted phase, the flattened free energy curve implies only a low further mechanical resistance of the DNA in agreement with previous studies [33]. Further increase in $\sigma$ may well result in increased absorption of unwinding stress in the melted region which eventually can lead to left-handed DNA structures.

## Modeling melting behavior upon unwinding by an Ising model

As a next step we compared the thermodynamic behavior extracted from our all-atom free energy simulations with predictions from an Ising model. In an Ising model one assumes that the stability of a dsDNA molecules is dominated by nearest neighbor interactions (base pairing and stacking interactions). In addition, the elastic twist deformation for the dsDNA is characterized by a single twist stiffness (persistence length) similar to the melted region that typically is represented by a much smaller twist stiffness. Due to its mathematically structure it is possible with an Ising model to calculate the complete partition function and all associated thermodynamic quantities of the dsDNA for varying unwinding stress $\sigma$.

Such models have been successfully employed in the past to predict DNA's denaturation probability [7, 14, 16, 17]. In contrast to the atomistic MD simulations, we use in the Ising model base-stacking and base-pairing parameters which have been determined experimentally [7, 49] (Table 3). Furthermore, we use an empirical bubble initiation parameter $\epsilon = 4.1$kcal/mol, which has been shown by Krueger et al. (2006) to reconcile base-stacking and pairing energies with basepair opening [7]. In analogy to the work of Benham [14], we set the stiffness constant of a denaturation bubble to an experimental estimate of $C = 0.79 cal/(mol \cdot deg^2)$, and describe its elastic energy by $E_{bubble} = C \cdot \frac{(\Delta \tau - 34.5 \cdot (n+1))^2}{2 \cdot n}$. Note, that the effective stiffness of the whole DNA depends on the number of molten base pairs n. For our Ising model calculations we use a twist persistence length $P = \frac{Contour \cdot k_{tw}}{k_B T}$ of the dsDNA region extracted from our unrestrained, all-atom MD simulations (by measuring the mean contour-length and the stiffness $k_{tw}$ as the inverse of the observed twist variance). We used here values extracted from the simulations because of a possible sequence dependence. However, we obtained a twist persistence length of 110.8 nm for the AT-sequence and 119.9 nm for the GC-sequence, respectively, indicating a rather small difference. Overall, the extracted twist persistence length is close to experimental estimates in the range of 100-120 nm [27] and close to what we obtained from previous simulations [33].

In the Ising model, we discretize base pairing by the eigenstates $|i>$, $|j> := \begin{pmatrix} 1 \\ 0 \end{pmatrix}$ for an intact base pair and $\begin{pmatrix} 0 \\ 1 \end{pmatrix}$ for a denatured base pair. The resulting Hamiltonian is then given in Eq 4. Here, the elements of the interaction matrix $H_{ij}$ are related to the state of the base pairs $<i|$ and $|j>$, e.g. the entry on the upper left of $H_{ij}$ describes the state where both base pairs are

**Table 3. Stacking energies of the ten different base pair steps used in the Ising model.** For base pairing we used $E_{bp}$ = 0.64 and $E_{bp}$ = 0.12 for AT and GC pairs, respectively. All values have been adapted from [49] and are in units of kcal/mol.

|             | ag    | ac    | at    | aa    | ta    |
|-------------|-------|-------|-------|-------|-------|
| $E_{stack}$ | -1.44 | -2.19 | -1.72 | -1.49 | -0.57 |
|             | tc    | tg    | gg    | gc    | cg    |
| $E_{stack}$ | -1.81 | -0.93 | -1.82 | -2.55 | -1.29 |

intact and is hence not penalized by any potential. By simple matrix multiplication one can calculate the nearest neighbor energy contribution of each state to the total energy of the system.

$$
H = \begin{cases} \Sigma_{i=0,j=i+1}^{N} \langle i| \begin{pmatrix} 0 & \epsilon - E_{bp}^{j} - E_{stack}^{ij} \\ -E_{bp}^{i} - E_{stack}^{ij} & -E_{bp}^{j} - E_{bp}^{i} - E_{stack}^{ij} \end{pmatrix} |j> +E_{bubble} & ,n>0 \\ \\ \dfrac{k_B TP}{2L}\Delta\tau^2 & ,n=0 \end{cases} \tag{5}
$$

The partition sum can then be obtained through a transfer-matrix calculation that sums over all possible states and the corresponding Boltzmann weight:

$$
Z = exp[-\beta E_{bubble}] \cdot <0 | \Pi_{i=0,j=i+1}^{N} exp[-\beta H_{ij}] | N+1> -exp[-\beta E_{bubble}] + exp[-\beta H_{n=0}] \tag{6}
$$

We set $<0| = |N+1> = \begin{pmatrix} 1 \\ 0 \end{pmatrix}$, as these two overhang base pairs are not

affected by torsional stress. Knowledge of the partition sum Z allows us to calculate all thermodynamic quantities and the dependence on torsional stress. The approach reflects the setup of our MD simulations. However, we must ensure that the number of denatured eigenstates equals n and hence only keep according terms in our implementation of the Ising model. For optimization purposes, we neglect macrostates containing more than one denaturation bubble. This approximation is justified, as our fragments are relatively short (compared to the size of plasmids) and such states are penalized by another $4.1 kcal/mol$. Obviously, the partition sum is a function of global unwinding $\Delta\tau$. The partition sum allows us to directly calculate the free energy profile vs. $\sigma$ given by $F = -k_B T \cdot ln(Z)$ which can be compared to the MD-based PMF calculations vs. $\sigma$. The resulting free energy profile is shown in Fig 9. It is in good qualitative agreement with the free energy profile obtained from our US-simulations (PMFs): Close to equilibrium, the Ising model prediction coincides very well with the MD-based PMFs. Remarkably, although based on different (empirical) parameters the Ising model also shows a flattening in the free energy profile upon transition to the melted phase. It is a consequence of the comparable low elastic constant of denaturation bubbles and hence reveals an enthalpic character of DNA denaturation. The Ising model, however, predicts a slightly earlier onset of phase transition than obtained from our all-atom free energy simulations. In our simulations, the intact phase is more stable by $\sim 2 kcal/mol$. One can identify 2 likely reasons: First, in our simulations we only find melting of the TATA-Box region, as bubble closing and reopening at a different position may occur on a too large timescale to be sampled in the current all-atom MD simulations. Thus, in this case our simulations entropically underestimate the denaturation phase. By means of the Ising model, we estimate this effect to correspond to only $\sim 0.40$ (AT), 0.09 (GC) $kcal/mol$. Second, we suppose base-stacking to be slightly overstabilizing in current DNA force-fields. This argument has also been made in previous studies [50, 51, 52].

Besides, the Ising model also allows us to determine local melting probabilities. Here, we obtain a dominant contribution from the TATA-Box and see that increased global C/G-content promotes earlier melting of this region (Fig 10). This phenomenon has already been found experimentally by Vlijm et al. (2015) [29]. We explain this finding by a melting competition between the segments of the DNA, as melting of one segment has a relaxation-contribution on all other segments and hence decreases their melting probability. Thus, we think of it as an entropic rather than an enthalpic phenomenon, as the persistence lengths of our sequences are also very similar. Furthermore, we predict the length of the denaturation bubble

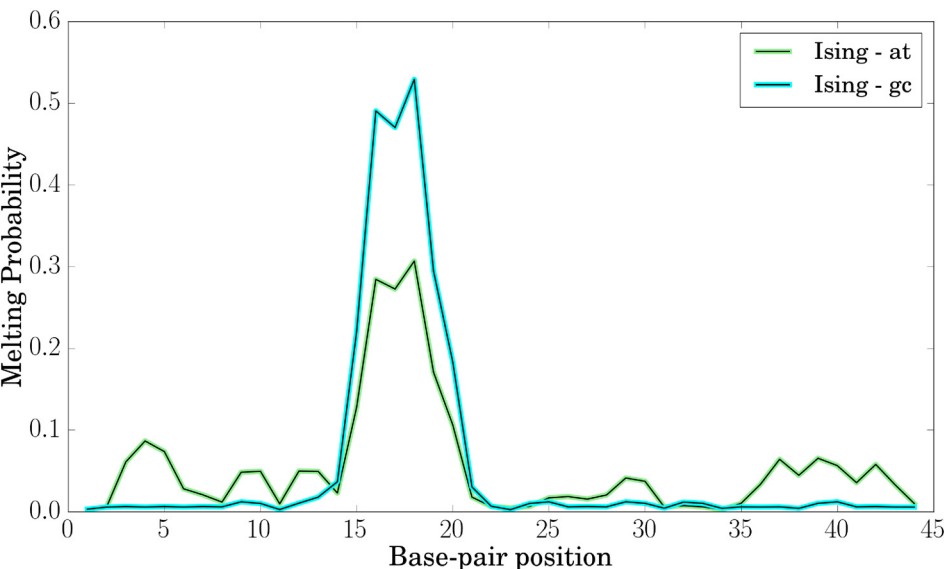

**Fig 10. Calculated melting probabilities along the sequence using the Ising model at a global supercoiling stress of $\sigma = 0.067$.**

to $\sim 1 - 3 bps$ (S32 Fig in S1 File) for our sequences. This also nicely resembles the observation from our simulations (Fig 3).

## Conclusions

While coarse-grained studies have already revealed localized melting in DNA [25, 35, 53], our study shows that specific localized melting of promoter-like TATA box segments is possible during all-atom MD simulations upon global unwinding. Most of the previous atomistic studies in this field have either focused on the harmonic regime in shorter duplexes [31, 32, 33] or on circular DNA [34].

Randall et al. (2009) conducted pioneering atomistic simulations also on larger sequences (up to 41 bps) under torsional stress, however, on significantly shorter time scales compared to the present study [18]. In these simulations the length of the DNA is constrained to fit exactly to the box dimension which also constrains (fixes) the total twist of the DNA to fit exactly into the periodic box (the DNA is periodically extended along the periodic simulation box). Although elegant for constraining the overall twist, it has the significant drawback that any lateral motion or bending of the DNA ends is completely suppressed. In the present study full azimuthal freedom (shrinking or extension of the DNA) and also some DNA bending to relax stress are possible representing a more realistic scenario. In atomistic MD simulations of Mitchell et al. [34] on supercoiled DNA minicircles the authors found formation of kinks, base pair opening and melting events.

However, neither of these studies provides a free energy analysis of the induced DNA melting or a comparison with an independent physical (Ising) model based entirely on empirical parameters nicely supporting the validity of the atomistic force field simulations. In addition to insights into the localized DNA melting process and associated energetic and structural changes our study demonstrates a significant redistribution (and fluctuation) of elastic deformation energy along the DNA molecule upon applying a global unwinding restraint. The gradual build-up of a non-uniform elastic energy distribution with increased unwinding is key for triggering the melting process. In relaxed DNA the elastic energy is evenly distributed

among the relevant conformational parameters. This changes quite dramatically upon inclusion of a global twist restraint and indicates how deformation energy can accumulate at certain DNA regions, a process that so far has neither been investigated experimentally nor theoretically in any systematic fashion.

Our calculations indicate a significant sequence dependence that may strongly influence the binding and recognition by proteins in supercoiled vs. linear unrestrained DNA. The non-uniform absorption of global stress can lead to a visible deformation of sequences which are then specifically recognized by proteins. In the intact non-melted phase, the TATA-boxes in both DNA molecules remained overall less torsionally stressed compared to the C/G flanking regions. Here, our results support previous notions of specific elements acting as 'twist capacitors' by means of transitions in the backbone conformation from BII to BI [32]. It is possible that overwinding evokes the opposite behavior and mainly stresses A/T segments by switching from BI to BII states, an issue that will be investigated in future studies.

In the MD simulations, the denaturation bubbles appear as undertwisted not-well structured regions allowing the rest of the sequences to relax towards near B-form. Experiments have shown that denaturation bubbles generated by global DNA unwinding can form a helical, left-handed structure [23, 24, 30]. We like to emphasize that for the local melting of segments within much longer DNA molecules (e.g. on the length-scale of kilo-bps) the denaturation bubbles also have to absorb a larger amount of unwinding which in turn can lead to formation of left-handed denaturation bubbles. Indeed, our simulations support this possibility because after local melting only little resistance to further global unwinding of this segment was observed (Fig 6).

In order to further check the validity of our simulation results and also to compare it with experimental data we designed an Ising model similar to the Benham model [1, 14, 15]. It allows us to directly compare the average free energy of the constrained system as well as the melting probability under torsional stress extracted from the simulations with the Ising model. The Ising model is based on empirical nearest neighbor base pair/stacking parameters as well as DNA twist elasticities.

A remarkably good agreement for the necessary torsional stress to induce a melting transition and for the position of the melting segment was obtained. In addition, the shape of the free energy vs. supercoiling density agrees excellently in the elastic regime and differs only in the phase stability and onset of phase transition. It emphasizes the overall quite realistic description by the force field but also indicates a slight over-stabilization of the dsDNA by the MD force field.

Our simulation study indicates that MD-simulations allow one to study in molecular detail the associated structural changes, the required free energy and the redistribution of elastic deformation energy upon global unwinding of DNA. In the future this can help to better understand how DNA supercoiling can modulate the binding properties of DNA for proteins and how it contributes to many important biological processes: Inhibition of the DNA gyrase enzymes, for instance, inhibits the repair of UV-damages in bacterial DNA [3] and Dittmore et al. (2017) have shown that supercoiling locates mismatches in DNA [2].

## Supporting information

**S1 File. This file contains information on the free energy simulations, calculation of stiffness and covariances, elastic energy, Ising model, calculation of geometric parameters and associated figures.**
(PDF)

## Acknowledgments

We thank P. Marlow for helpful discussions and acknowledge support by the Leibniz super computer center.

## Author Contributions

**Conceptualization:** Martin Zacharias.

**Data curation:** Korbinian Liebl.

**Formal analysis:** Korbinian Liebl.

**Funding acquisition:** Martin Zacharias.

**Investigation:** Korbinian Liebl.

**Methodology:** Korbinian Liebl.

**Project administration:** Martin Zacharias.

**Software:** Korbinian Liebl.

**Supervision:** Martin Zacharias.

**Validation:** Korbinian Liebl, Martin Zacharias.

**Writing – original draft:** Korbinian Liebl, Martin Zacharias.

**Writing – review & editing:** Martin Zacharias.

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
