## [Decision Letter · Decision Letter 0]

1 Apr 2020

PONE-D-20-05578

How global DNA unwinding causes non-uniform stress distribution and melting of DNA

PLOS ONE

Dear Dr. Zacharias,

Thank you for submitting your manuscript to PLOS ONE. After careful consideration, we feel that it has merit but does not fully meet PLOS ONE’s publication criteria as it currently stands. Therefore, we invite you to submit a revised version of the manuscript that addresses the all the points raised by both reviewers.

We would appreciate receiving your revised manuscript by May 16 2020 11:59PM. To enhance the reproducibility of your results, we recommend that if applicable you deposit your laboratory protocols in protocols.io, where a protocol can be assigned its own identifier (DOI) such that it can be cited independently in the future. For instructions see: http://journals.plos.org/plosone/s/submission-guidelines#loc-laboratory-protocols

We look forward to receiving your revised manuscript.

Kind regards,

Claudio M. Soares, Ph.D

Academic Editor

PLOS ONE

Journal Requirements:

Reviewers' comments:

Reviewer's Responses to Questions

**Comments to the Author**

1. Is the manuscript technically sound, and do the data support the conclusions?

Reviewer #1: Yes

Reviewer #2: Yes

2. Has the statistical analysis been performed appropriately and rigorously? 

Reviewer #1: Yes

Reviewer #2: Yes

3. Have the authors made all data underlying the findings in their manuscript fully available?

Reviewer #1: Yes

Reviewer #2: Yes

4. Is the manuscript presented in an intelligible fashion and written in standard English?

Reviewer #1: Yes

Reviewer #2: Yes

5. Review Comments to the Author

Reviewer #1: In this paper the authors present a Molecular Dynamics (MD) work to characterize the structural changes and energetics during the unwinding and melting of DNA duplexes containing a TATA-box transcription start sequence. The authors find that the elastic deformation of two DNA constructs is distributed non-uniformly during unwinding and results in the local melting of the TATA-box, which also absorbs the deformation energy of the duplex while the rest of the sequence relaxes to B-form. The authors find good agreement between the free-energy profile reconstructed along the degree of unwinding from fully-atomistic MD and from an Ising model based on experimental parameters, thus corroborating the use of MD simulations to study the molecular details and energetics of global DNA unwinding processes. Overall the study is interesting, clearly written, and should be published after the authors have addressed the following minor issues noticed:

1) Can the authors demonstrate that, or at least argue on, the local melting of the TATA-box during global unwinding is independent on its positioning along the duplex? What if the TATA-box is positioned, for example, between base pairs 10-15 or 20-25 ? Would the TATA-box continue to absorb the majority of the deformation energy and locally melt (and not other regions) at sigma around 0.073 ?

2) It is now evident that local base-pair melting is a step-wise asymmetric process with one of the two Watson-Crick bases having higher probability to fray than the complementary one (PNAS 100, 13922–13927 (2003); J Phys Chem B 113, 2614–2623 (2009); J Comput Chem 32, 3354–3361 (2011); PNAS 113, 116–121 (2016); J Mol Model. 23, 226 (2017); PNAS 116, 22471-22477 (2019)). In this context, it would be interesting for the authors to comment on the molecular details governing the melting of the TATA-box from such perspective. Did they observe, for instance, preferential fraying of Thymine(T) leading to melting of the TATA-box ?

Reviewer #2: This well-written manuscript describes a computational study of double stranded DNA under torsional stress. Specifically, the authors performed all-atom molecular dynamics simulations of two 50-basepair DNA fragments different in their CG content but featuring the same TATA box, a fragment of DNA that is frequently present at the start of DNA transcription sites. The under-twisting of DNA was realized by means of torsional restraints. Gradual increase of the restraints allowed the authors to observe redistribution of the mechanical energy at the bp level, melting of the TATA box, and, impressively, obtain the free-energy costs associated with the above transitions. The authors rationalize the results of the simulations using an Ising model.

This is a very impressive study that has the potential to become a classic paper in the field of computational DNA mechanics. The observations of TATA box melting and redistribution of mechanical deformation energy are particularly valuable. The manuscript is clearly written and can be followed by a person familiar with the subject. The manuscript can be published in its present form, though it would be nice if the authors could split the very long first paragraph of the Conclusions into several shorter paragraphs.

6. PLOS authors have the option to publish the peer review history of their article (what does this mean?). If published, this will include your full peer review and any attached files.

Reviewer #1: No

Reviewer #2: Yes: Aleksei Aksimentiev

---

## [Author Response · Author response to Decision Letter 0]

22 Apr 2020

Garching, 12.4.2020

Dear Dr. Soares

Thank you for returning our manuscript entitled “How global DNA unwinding causes non-uniform stress distribution and melting of DNA" by K. Liebl and M. Zacharias and the comments of the reviewers. In the following we like to comment on the concerns of the reviewers and indicate the changes and additions we have made to the manuscript. We also provide a version of the manuscript with all changes marked red. 

Referee: 1

Reviewer #1: In this paper the authors present a Molecular Dynamics (MD) work to characterize the structural changes and energetics during the unwinding and melting of DNA duplexes containing a TATA-box transcription start sequence. The authors find that the elastic deformation of two DNA constructs is distributed non-uniformly during unwinding and results in the local melting of the TATA-box, which also absorbs the deformation energy of the duplex while the rest of the sequence relaxes to B-form. The authors find good agreement between the free-energy profile reconstructed along the degree of unwinding from fully-atomistic MD and from an Ising model based on experimental parameters, thus corroborating the use of MD simulations to study the molecular details and energetics of global DNA unwinding processes. Overall the study is interesting, clearly written, and should be published after the authors have addressed the following minor issues noticed:

Response: We thank the reviewer for the encouraging comment.

1) Can the authors demonstrate that, or at least argue on, the local melting of the TATA-box during global unwinding is independent on its positioning along the duplex? What if the TATA-box is positioned, for example, between base pairs 10-15 or 20-25 ? Would the TATA-box continue to absorb the majority of the deformation energy and locally melt (and not other regions) at sigma around 0.073 ?

Response: The torsional deformation on the DNA is introduced gradually in the simulations that are much longer than any prior simulations that introduce torsion stress in DNA. The fact that the calculated free energy change for the melting process is well converged indicates that sufficient time is given for the distribution of the stress along the DNA. Secondly, in both DNA sequences we used for the simulations the TATA-box segment was located off center. Nevertheless, melting occurred in both cases at the TATA-box. These two observations strongly suggest that the effect (as expected) is not depending on the position of the TATA-box in the DNA. 

2) It is now evident that local base-pair melting is a step-wise asymmetric process with one of the two Watson-Crick bases having higher probability to fray than the complementary one (PNAS 100, 13922–13927 (2003); J Phys Chem B 113, 2614–2623 (2009); J Comput Chem 32, 3354–3361 (2011); PNAS 113, 116–121 (2016); J Mol Model. 23, 226 (2017); PNAS 116, 22471-22477 (2019)). In this context, it would be interesting for the authors to comment on the molecular details governing the melting of the TATA-box from such perspective. Did they observe, for instance, preferential fraying of Thymine(T) leading to melting of the TATA-box ?

Response: We checked the simulation trajectories for the step-wise asymmetry. We found indeed that upon melting thymine bases showed a higher tendency for fraying by a factor of 1.2 to 1.3 compared to adenine in the TATA-box. The paragraph on page 9 was modified accordingly and new references were added.

Reviewer #2: This well-written manuscript describes a computational study of double stranded DNA under torsional stress. Specifically, the authors performed all-atom molecular dynamics simulations of two 50-basepair DNA fragments different in their CG content but featuring the same TATA box, a fragment of DNA that is frequently present at the start of DNA transcription sites. The under-twisting of DNA was realized by means of torsional restraints. Gradual increase of the restraints allowed the authors to observe redistribution of the mechanical energy at the bp level, melting of the TATA box, and, impressively, obtain the free-energy costs associated with the above transitions. The authors rationalize the results of the simulations using an Ising model.

This is a very impressive study that has the potential to become a classic paper in the field of computational DNA mechanics. The observations of TATA box melting and redistribution of mechanical deformation energy are particularly valuable. The manuscript is clearly written and can be followed by a person familiar with the subject. The manuscript can be published in its present form, though it would be nice if the authors could split the very long first paragraph of the Conclusions into several shorter paragraphs.

Response: We thank the reviewer for the encouraging statement. We followed the recommendation of the reviewer and split the Conclusion section into paragraphs to improve readability of this section.

Finally, we like to thank the reviewers for the fair comments and hope that with the additions and changes we have made to the manuscript it is now acceptable for publication in PlosOne.

Yours sincerely,

Korbinian Liebl, Martin Zacharias

---

## [Decision Letter · Decision Letter 1]

27 Apr 2020

How global DNA unwinding causes non-uniform stress distribution and melting of DNA

PONE-D-20-05578R1

Dear Dr. Zacharias,

We are pleased to inform you that your manuscript has been judged scientifically suitable for publication and will be formally accepted for publication once it complies with all outstanding technical requirements.

With kind regards,

Claudio M. Soares, Ph.D

Academic Editor

PLOS ONE

Additional Editor Comments (optional):

Reviewers' comments:

Reviewer's Responses to Questions

**Comments to the Author**

1. If the authors have adequately addressed your comments raised in a previous round of review and you feel that this manuscript is now acceptable for publication, you may indicate that here to bypass the “Comments to the Author” section, enter your conflict of interest statement in the “Confidential to Editor” section, and submit your "Accept" recommendation.

Reviewer #1: All comments have been addressed

2. Is the manuscript technically sound, and do the data support the conclusions?

Reviewer #1: Yes

3. Has the statistical analysis been performed appropriately and rigorously? 

Reviewer #1: Yes

4. Have the authors made all data underlying the findings in their manuscript fully available?

Reviewer #1: Yes

5. Is the manuscript presented in an intelligible fashion and written in standard English?

Reviewer #1: Yes

6. Review Comments to the Author

Reviewer #1: I am glad to confirm that the Authors have sufficiently addressed my concerns with their revisions.

7. PLOS authors have the option to publish the peer review history of their article (what does this mean?). If published, this will include your full peer review and any attached files.

Reviewer #1: No

---

## [Editor Report · Acceptance letter]

1 May 2020

PONE-D-20-05578R1 

How global DNA unwinding causes non-uniform stress distribution and melting of DNA 

Dear Dr. Zacharias:

I am pleased to inform you that your manuscript has been deemed suitable for publication in PLOS ONE. Congratulations! Your manuscript is now with our production department. 

With kind regards,

on behalf of

Dr. Claudio M. Soares 

Academic Editor

PLOS ONE